# Experimental Study on Colpitts Chaotic Oscillator-Based Communication System Application for the Internet of Things

**Darja Cirjulina** [1,*] , **Ruslans Babajans** [1] , **Filips Capligins** [1] , **Deniss Kolosovs** [1] **and Anna Litvinenko** [2]

1 Institute of Microwave Engineering and Electronics, Riga Technical University, 6A Kipsalas Street, LV-1048 Riga, Latvia; ruslans.babajans@rtu.lv (R.B.); filips.capligins@rtu.lv (F.C.); deniss.kolosovs@rtu.lv (D.K.)
2 SpacESPro Lab, Riga Technical University, 6A Kipsalas Street, LV-1048 Riga, Latvia; anna.litvinenko@rtu.lv
* Correspondence: darja.cirjulina@rtu.lv

**Abstract:** This manuscript presents an experimental study of Quadrature Chaos Shift Keying (QCSK) as a means to tighten the physical layer security of Internet of Things (IoT) communication. Our study examines the characteristics and operational aspects of chaos oscillators, prioritizing low-power functionality, resilient chaotic oscillations, and resistance to parameter variations and noise. This study emphasizes the key role of chaos oscillators in enhancing IoT security, showcasing their potential to ensure data integrity. The findings elucidate the dynamics and synchronization stability of the selected oscillators, providing insights into their suitability for secure communication systems. This comprehensive analysis contributes to advancing secure communication methodologies for the expanding landscape of wireless sensor networks in the Internet of Things, underscoring the significance of chaos oscillators in ensuring robust and secure data transmission.

**Keywords:** nonlinear systems; chaos shift keying; chaos oscillator; chaotic synchronization; communication system; signal processing; internet of things

## 1. Introduction

Our society is increasingly witnessing the integration of diverse smart environments driven by Internet of Things (IoT) solutions. The concepts of smart cities, streamlined processes in smart environments, and efficient patient monitoring and treatment in smart healthcare are frequently discussed. According to the Ericsson Mobility Report [1], IoT connections are projected to rise to 38.9 billion by 2029, a significant increase from the current 15.7 billion. The number of IoT connections will more than double in the next five years. These expansive networks manage substantial volumes of sensitive data, rendering them attractive targets for cyber threats [2], thus necessitating tightening security measures in such networks. The proliferation of IoT organizations and alliances attempting to enforce diverse protocols and security measures contributes to the complexity and credibility challenges in IoT security [3]. Researchers are actively developing innovative solutions to bolster IoT security despite these obstacles. As highlighted in [4], the primary research domains include blockchain, fog computing, machine learning (ML), edge computing, and encryption-based solutions.

Blockchain-based security operates as a distributed ledger with chronological, time-stamped entries. Each entry is linked to the previous one using a cryptographic hash function, and individual transactions are stored in a Merkle tree [4,5]. However, the decentralized nature required by blockchain poses significant challenges in IoT networks due to resource limitations in the majority of nodes. Implementing extensive blockchain-based security, including address management, communications, and distribution, becomes complex in such environments [6].

Cloud computing can address a limited scope of IoT issues, leading to Cisco's introduction of fog computing in 2012 to supplement its capabilities [4]. While cloud stor-

age boasts accessibility advantages, it faces the considerable drawback of vulnerability to data breaches, especially due to its centralized nature. Fog computing represents a middle-ground solution, leveraging nodes closer to end devices. This setup facilitates data processing at these intermediary nodes, reducing the amount of traffic between devices and the cloud. Although this setup does offer advantages by addressing some latency issues, it does not entirely resolve vulnerabilities between the end devices and the fog layer [7]. Edge computing introduces a significant shift by moving most processing to end devices. This approach significantly improves latency by localizing data processing but necessitates substantially higher resource allocation and computational power. However, this presents a security concern, as all data resides on these end-user devices, making the network vulnerable to misconfigurations and potential security breaches [8].

ML offers the capability to detect intrusions and anomalies in IoT networks. However, its widespread use faces limitations, including the selection and training of appropriate ML algorithms and the complex task of preparing IoT-generated data for accurate predictions. The abundance of data from IoT devices, often riddled with outliers and missing values, poses challenges for ML efficiency, demanding meticulous cleaning and preprocessing [4,9,10].

Encryption plays a vital role in the IoT, though it does not prevent security risks. Implementing link-level symmetric-key encryption like DES or AES ensures secure transmission; however, in this case, the system requires key distribution mechanisms. That can be implemented by applying asymmetric-key approaches such as RSA or elliptic-curve-based cryptography, yet managing keys across vast networks is complex, especially in decentralized settings. IoT devices, often resource-constrained and real-time-reliant, need help with traditional encryption's demands [11]. Lightweight cryptography offers a simpler alternative but requires new algorithms to balance simplicity with robustness. More robust encryption may pose risks as technology progresses, necessitating ongoing security development against evolving threats [12].

Security mechanisms operating at higher layers might not address attacks at the physical layer [13]. In IoT applications, considering the large number of devices and the possible alteration of communication by an adversary, an attack like a man-in-the-middle is easy to perform. While data theft might be prevented, interference in information transmission through broadcast signals could disrupt critical applications like vehicle-to-vehicle communication or medical IoT. Improving physical layer security addresses these concerns but is complex due to limited device resources. Hence, developing efficient and secure physical layer protection methods is a focal point in IoT security [5,11,14].

The chaos phenomenon presents a promising avenue for strengthening physical layer security in IoT systems. It offers diverse attributes such as a wide spectrum, noise-like waveform, narrow cross-correlation function, synchronization capability, and sensitivity to initial conditions while maintaining system determinism [13]. Integrating chaotic signals with coherent modulation techniques enhances security in IoT systems [15]. These signals find applications in fields like random number generation [16,17], encryption [18–21], radar systems [22–24], and IoT communications [25–27]. The versatility of chaotic signals in generating analog signals and digital logistic maps drives interest, particularly in FPGA-based image encryption using chaotic logistic maps [28–30]. Such systems provide secure communication channels for transmitting images across IoT sensing systems to domains like smart healthcare [31,32], where security and privacy are crucial. Ongoing efforts involve developing and studying communication systems based on analog chaos oscillators [33–35].

Simpler analog chaos oscillators can replace digital chaotic communication for resource-constrained IoT sensors and nodes. Various chaotic oscillators, including Colpitts [36], have been extensively studied [37–41]. Colpitts chaos oscillators, in particular, have garnered attention for their simple schematic. L.O. Chua [42,43] specified that a chaos oscillator must have at least three state variable elements (inductor, capacitor) to accumulate energy, an active element to support fluctuation, and a nonlinear element to ensure system nonlinear dynamics. The Colpitts chaos oscillator, comprising a transistor as both the

active and nonlinear element, along with an inductor and capacitors, allows for a more straightforward configuration of the fundamental frequency by adjusting reactive element values [44–47]. Bendecheche et al. developed a Colpitts chaos oscillator operating in the frequency range of 500 MHz to 15 GHz [48]. Due to its specific autocorrelation function shape, it finds application in UWB radar detection systems [22,23]. Moreover, that system can effectively utilize a wide spectrum, providing access to different spectral channels and avoiding transmitter detection. Researchers have also explored Colpitts chaos oscillator signals' numerical, statistical, and informational parameters [49–52], conducting Lyapunov exponent analysis to identify parameter regions resulting in chaotic behavior [53].

The simplicity of the Colpitts chaos oscillator circuit has made it a notable choice for data transmission. Rubezic et al. were the first to showcase Colpitts chaos oscillator synchronization in a drive-response system, enabling the development of a coherent communication system [54]. Luke T. Harwood et al. introduced binary phase-shift keying based on chaotic oscillators [55], while P. Canyelles-Pericas et al. suggested chaotic modulation using a schematic power supply [56]. Additionally, Ruslan Fattakhov et al. explored and studied security enhancements in OFDM systems by employing Discrete-nonlinear Colpitts oscillator-based communication [57]. A lot of different communication systems have been discussed; nevertheless, there are no proposals for enhancing IoT physical layer security by employing the Colpitts chaos oscillator.

This study introduces the Quadrature Chaos Shift Keying (QCSK) communication system based on the Colpitts chaos oscillator for IoT applications. The system employs two chaotic signals: shift keying by a bit sequence, synchronization using the substitution method, and coherent detection [58]. As the QCSK communication system is built using an analog chaos oscillator, as the first step of this study, it is essential to analyze its dynamics' stability. The main characteristics of the chaos oscillator—the ability to synchronize and low signal cross-correlation—depend on the nominal values of individual electronic components—and the system's synchronization relies on the similarity of chaos oscillators. Studying the Colpitts chaos oscillator model in both simulation and hardware is essential to ensuring the oscillator operates in chaotic mode. Depending on the initial conditions, a chaos oscillator can generate even periodic signals, but such signals would not ensure security in the communication system. Chaos oscillators' dynamic stability, synchronization durability, and frequency domain properties significantly impact the performance of the QCSK transmission system. In the current study, extensive synchronization analysis has been performed due to the precise signal detection dependency on QCSK system synchronization.

In previous works, frequency-modulated chaos shift keying (FM-CSK) and quadrature chaos phase-shift keying (QCPSK) communication systems were proposed [59,60] for IoT physical layer security enhancements. Still, the proposed systems had a significant shortcoming in the form of low data transmission rates. In this study, we address this shortcoming by increasing the fundamental frequency of the Colpitts chaos oscillator to increase the data transmission speed for IoT applications. The commonly used IoT protocol, the Narrowband Internet of Things (NB-IoT), can achieve transmission speeds up to 128 Kbps [61].

This manuscript consists of 5 sections. Section 2 explains the concepts and structure of the proposed QCSK communication system. In Section 3, the study on the Colpitts chaos oscillator is presented and examined both in simulation and hardware. Section 4 presents the QCSK communication system performance analysis in the AWGN channel. Section 5 concludes this paper.

## 2. Quadrature Chaos Shift Keying Communication System

The current section introduces a communication system concept established on quadrature modulation and chaos shift keying (CSK) principles. Figure 1 illustrates the block diagram outlining the proposed Quadrature Chaos Shift Keying (QCSK) communication system. The system's core is an analog chaos oscillator operating in a drive-response

configuration, resulting in a coherent communication system. In the transmitter, the drive oscillator generates three state variables ($X$, $Y$, and $Z$): voltages across capacitors and currents through inductors, as defined by circuit theory. On the receiver end, the response oscillator produces only two state variables ($X'$, $Y'$), with one variable ($Z'$) being replaced by the transmitted signal derived from the drive oscillator ($Z$).

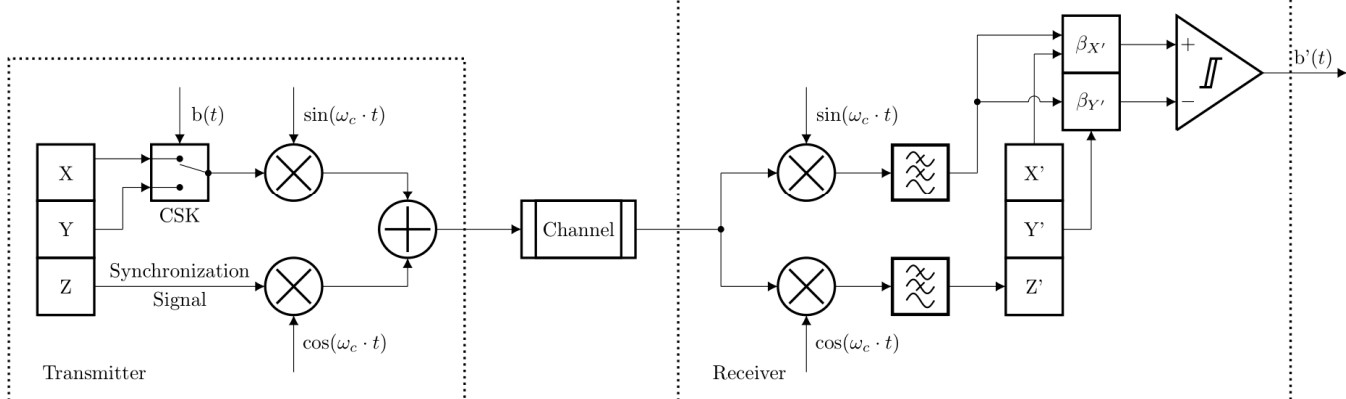

**Figure 1.** Quadrature Chaos Shift Keying (QCSK) communication system block-scheme, where $X$, $Y$, and $Z$ are drive oscillator state variables, $X'$, $Y'$, and $Z'$ are response oscillator state variables, b($t$) is binary information signal, CSK is binary information-signal-controlled switch that performs chaos shift keying, and b$'$($t$) is recovered binary information signal.

Within this configuration, the signal from the state variable ($Z$) of the drive circuit serves as a synchronization signal. The remaining state variables ($X$ and $Y$) of the drive circuit are utilized for information transmission (depicted in Figure 1 in the CSK block). Notably, all state variables represent chaotic signals, expressed as voltages across and currents through the physically implemented chaos oscillators' circuit elements.

The information-carrying signal undergoes formation via a binary information-signal-controlled switch, alternating between $X$ and $Y$ signals, where bit '1' corresponds to $X$ and bit '0' to $Y$. Prior to forming the information-carrying signal, DC-cancellation of $X$ and $Y$ signals is essential, ensuring matched amplitudes. Simultaneously, the DC component is removed from the synchronization signal $Z$. Both the formed information-carrying and synchronization signals subsequently undergo quadrature modulation and are transmitted through an Additive White Gaussian Noise (AWGN) channel.

Upon receiving, the signal undergoes quadrature demodulation and passage through a low-pass filter. Detecting information bits relies on calculating the correlation coefficient $\beta$ between the received signal and the $X'$ and $Y'$ signals from the response circuit. Detection of data bit '1' occurs if $\beta_{X'} > \beta_{Y'}$; otherwise, '0'. Accurate bit detection necessitates signal agreement between the drive and response circuits. Information detection relies on correlation coefficient calculations, highlighting the significance of the chaos signal's frequency. Ensuring a minimum of a few oscillations per bit becomes crucial for enabling accurate detection through correlation coefficient calculations. Moreover, this factor significantly impacts the system's noise immunity. Hence, chaotic synchronization between the two chaos oscillators is imperative for precise signal detection. Consequently, the subsequent section investigates the properties of chaos oscillators. Additionally, Section 4 performs Bit Error Ratio (BER) estimation, validates results, and assesses the communication system's noise resilience.

## 3. Colpitts Chaos Oscillator

This section studies the nonlinear dynamics of the Colpitts chaotic oscillator, which is distinguished by its noise-like characteristics, sensitivity to initial conditions, and sharp signal cross-correlation function. Furthermore, this study delves into the synchronization

aspects within drive-response systems. The Colpitts chaotic oscillator was chosen for this study because of its simple design and capacity to adjust the fundamental frequency. The primary objective is to evaluate the robustness of a Colpitts chaos oscillator drive-response system in simulation and prototype.

### 3.1. Colpitts Chaos Oscillator Mathematical Model

According to L.O. Chua et al. [42,43], to produce a chaotic signal, an oscillator must have at least three state variable elements, such as capacitors or inductors, a nonlinear element, such as a diode or transistor, and an active component that is critical for maintaining oscillations. The Colpitts chaos oscillator design includes an NPN bipolar junction transistor $Q_1$, biased within its active range via $V_1$, $R_1$, and $V_2$. The feedback is incorporated via inductor $L_1$ with series resistance $R_L$ and a capacitive divider consisting of $C_1$ and $C_2$ [36]. Figure 2 depicts the Colpitts chaotic oscillator circuit.

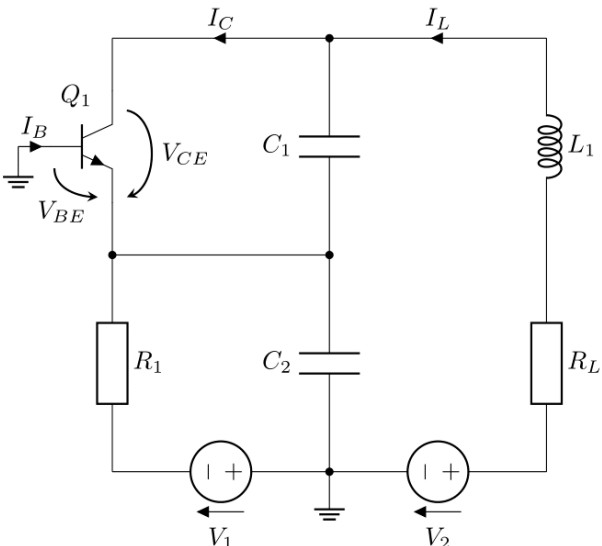

**Figure 2.** Colpitts Chaos oscillator circuit, where $C_1$ = 4.5 nF, $C_2$ = 4.5 nF, $L_1$ = 10 μH, $R_L$ = 35 Ω, $R_1$ = 400 Ω, $V_1$ = 5 V, $V_2$ = 5 V, $Q_1$ is NPN bipolar junction transistor.

The mathematical model of the Colpitts chaos oscillator is described with the following equations:

$$\begin{cases} C_1 \frac{\mathrm{d}V_{CE}}{\mathrm{d}t} = I_L - I_C \\ C_2 \frac{\mathrm{d}V_{BE}}{\mathrm{d}t} = \frac{V_1 + V_{BE}}{R_1} - I_L - I_B \\ L_1 \frac{\mathrm{d}I_L}{\mathrm{d}t} = V_2 - V_{CE} + V_{BE} - I_L \cdot R_L \end{cases}, \tag{1}$$

where $V_{CE}$ is collector-emitter voltage, $V_{BE}$ is base-emitter voltage, $V_1$ and $V_2$ are circuit input voltages, $I_L$ is inductor current, $I_C$ is collector current, and $I_B$ is base current.

The nonlinearity of the Colpitts chaos oscillator is introduced in the base current ($I_B$) of transistor $Q_1$. which is defined as follows:

$$I_B = \begin{cases} 0, & \text{if } V_{BE} \leq V_{TH} \\ \frac{V_{BE} - V_{TH}}{R_{ON}}, & \text{if } V_{BE} > V_{TH} \end{cases}, \tag{2}$$

$$I_C = \beta_F \cdot I_B, \tag{3}$$

where $V_{TH}$ is the threshold voltage, $R_{ON}$ is the small-signal on-resistance of the base-emitter, and $\beta_F$ is the forward current gain of the transistor.

### 3.2. Chaotic Synchronization

Establishing chaotic synchronization between chaos oscillators within the transmitter and receiver is crucial to ensuring secure communication in coherent data transmission systems. Pecora and Carroll introduced a synchronization method [58] that involves substituting the signal within the response circuit with a signal sourced from the drive circuit. This approach facilitates information recovery from the received signal, constituting a vital aspect of the synchronization process.

The synchronization via substitution method in the Colpitts chaos oscillator can be established by utilizing voltage across capacitors $C_1$ or $C_2$. The drive oscillator generates three state variables ($V_{C1}$, $V_{C2}$, and $I_L$). After the synchronization, the response oscillator generates only two state variables. If $V_{C2}$ is used for synchronization, the response chaos oscillator generates $V'_{C1}$ and $I'_{L1}$. In other solutions, the response chaos oscillator generates $V'_{C2}$ and $I'_L$.

An example of the synchronization in the chaos oscillator is depicted in Figure 3. The signal $V_{C1}$ from the drive oscillator replaces the voltage across capacitor $C'_1$ in the response oscillator. Initially, for the first two milliseconds, both the drive and response chaos oscillators operate independently. However, after 2 ms, the drive-response synchronization process is complete, enabling the response oscillator to replicate the behavior of the drive oscillator.

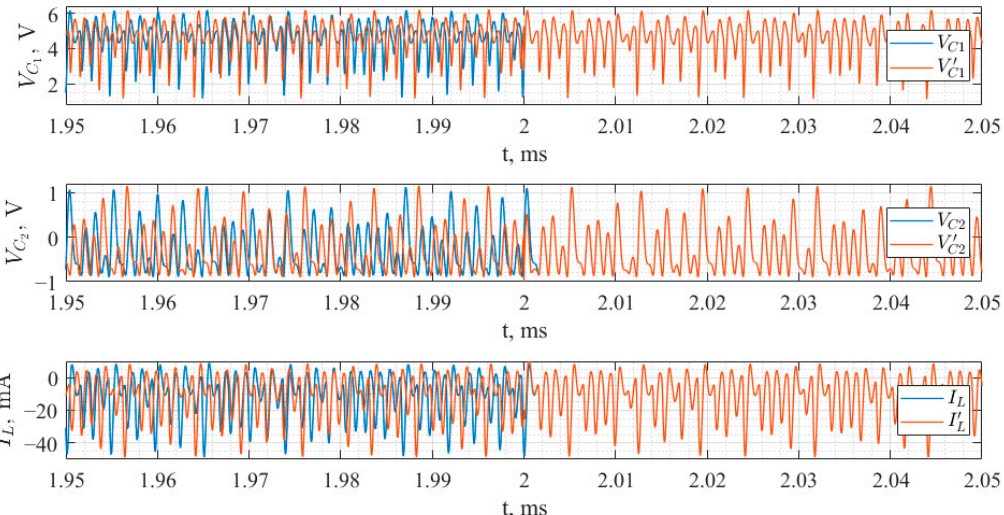

**Figure 3.** Colpitts Chaos oscillator drive-response synchronization, where $V_{C1}$, $V_{C2}$, and $I_L$ represent signals in drive circuit and $V'_{C1}$, $V'_{C2}$, and $I'_L$ represent signals in response circuit.

The synchronization between the drive and response circuits of chaotic oscillators must be evaluated. Pearson's correlation coefficient $\beta$ was chosen for this purpose since it can evaluate how similar the signals are on a scale of $-1$ to 1. The Pearson's correlation coefficient $\beta$ is calculated as follows:

$$\beta = \frac{\sum_{i=1}^{n}(x_i - \overline{x})(y_i - \overline{y})}{\sqrt{\sum_{i=1}^{n}(x_i - \overline{x})^2}\sqrt{\sum_{i=1}^{n}(y_i - \overline{y})^2}} \tag{4}$$

where $x$ represents the drive circuit signal and $y$ represents the identical element signal in the response circuit. When signals match identically, $\beta$ equals 1. In the case of identical signals but with opposite phases, $\beta$ takes a value of $-1$. For signals that differ and are dissimilar, $\beta$ equals 0.

The correlation coefficient between the voltages across their respective components converges toward one when the circuits achieve synchronization. This significant correlation indicates that the drive-response system's two chaotic oscillators are well-synchronized.

An operational amplifier working in voltage-follower mode establishes the implementation of chaotic synchronization, enabling data transmission.

### 3.3. Study on the Colpitts Chaos Oscillator in a Simulation Environment and Prototype

The chaotic system is characterized by sensitivity to the initial conditions. In the case of the analog circuit, the sensitivity to the electronic component's nominal values and parasitic characteristics can be considered an implication of this property. Even for identical initial conditions, the slight deviations in nominal values imply changes in the system's state after a short time interval. Simulation and experimental studies aim to determine the Colpitts chaos oscillator parameters that provide chaotic dynamics for the fundamental frequency of 96.86 kHz.

The simulations were run in the LTspice simulation environment, and the prototype measurements were taken with Analog Discovery Pro. MATLAB was used to process the signals for both the simulation and the prototype. The first stage of model verification involved comparing the attractors' two-dimensional projections to those reported in the original publication [36]. Following that, the Z1TEST was used to evaluate the system's signal behavior, quantifying it on a scale of 0 to 1 [62]. In this case, 0 indicates periodic behavior, while 1 represents chaotic behavior.

The Z1TEST consists of four primary steps. The first step is calculating the Fourier decomposition of the input signal $X(n)$. The next step is determining the mean square displacement $M_c(n)$ using the Fourier decomposition results. The final stage involves determining the correlation coefficient $K_c$ between $M_c(n)$ and the linearly growing array. For precise results, this procedure is conducted several times. The Z1TEST returns the median of all estimated $K_c$ values.

The nominal values of the reactive components were reduced to achieve a higher fundamental frequency for the Colpitts chaotic oscillator. Modifications were made to the original working frequency of 96.86 kHz, resulting in an increased fundamental frequency of 968.59 kHz. This improvement was achieved by changing the reactive component values to $L_1 = 10$ µH and $C_1 = C_2 = 5.4$ nF. Thomson's formula was used to compute the fundamental frequency, which considered the equivalent capacitance produced from the series combination of $C_1$ and $C_2$. The change is critical because it provides the path for faster data transfer speeds inside the QCSK communication system.

### 3.3.1. Simulation Study

Following modifications ($L_1 = 10$ µH and $C_1 = C_2 = 5.4$ nF), the Colpitts chaotic oscillator model was created in the LTspice simulation environment. Resistor nominal and supply voltage values were used the same as in the original study [36], as well as the 2N2222 NPN bipolar junction transistor. The simulation lasted 500 µs and used a maximum timestep of 0.1 µs. Signals generated in the LTspice simulation environment were exported to text files and processed in MATLAB. Following the simulation, the two-dimensional projections of the resulting attractor, as shown in Figure 4, were compared to those from the original study [36]. The observed congruence between the projections indicates that the LTspice model was successfully created. The Z1TEST result was 0.99, suggesting that the system's behavior is chaotic.

The next goal was to synchronize two oscillators in the drive-response system. The model of the Colpitts chaos oscillator in the LTspice simulation environment was expanded—the second identical Colpitts chaos oscillator circuit was added. The synchronization was established through an operational amplifier in voltage-follower mode. After the simulation, signals were exported and processed in MATLAB. The correlation coefficient between corresponding nodes in the circuit was used to evaluate synchronization. This assessment was used to determine the level of synchronization attained. Research revealed that synchronization via the substitution method in Colpitt's chaos oscillator can be established by utilizing voltage across capacitors $C_1$ or $C_2$. The correlation coefficient values in both configurations reached a notable 0.98, proofing the system's reliable synchronization.

Further insights into the system dynamics were obtained by analyzing signal spectra, contributing to an enhanced understanding of its behavior. Figure 5 presents the $V_{C1}$, $V_{C2}$, and $I_{L1}$ signal spectra. A careful examination of the signal spectra reveals that the signal band is mostly concentrated around the fundamental frequency of the chaos oscillator. It is also worth noting that the signals have a wide band with gradually decreasing amplitudes. This specific signal feature demands careful consideration while calculating filters for the communication system. Understanding these spectral characteristics is critical for enhancing system performance and guaranteeing efficient signal processing.

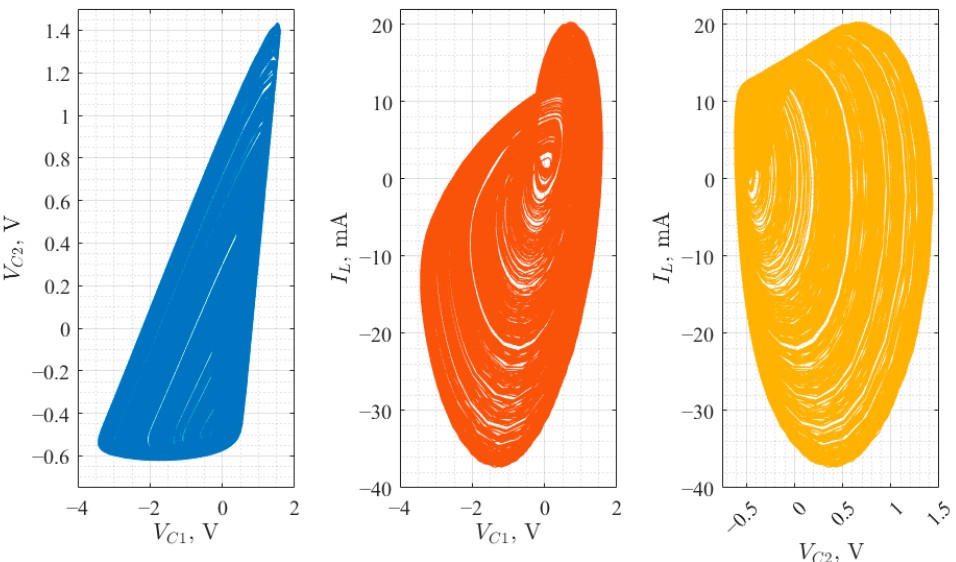

**Figure 4.** Colpitts Chaos oscillator attractor two-dimensional projections in LTspice simulation.

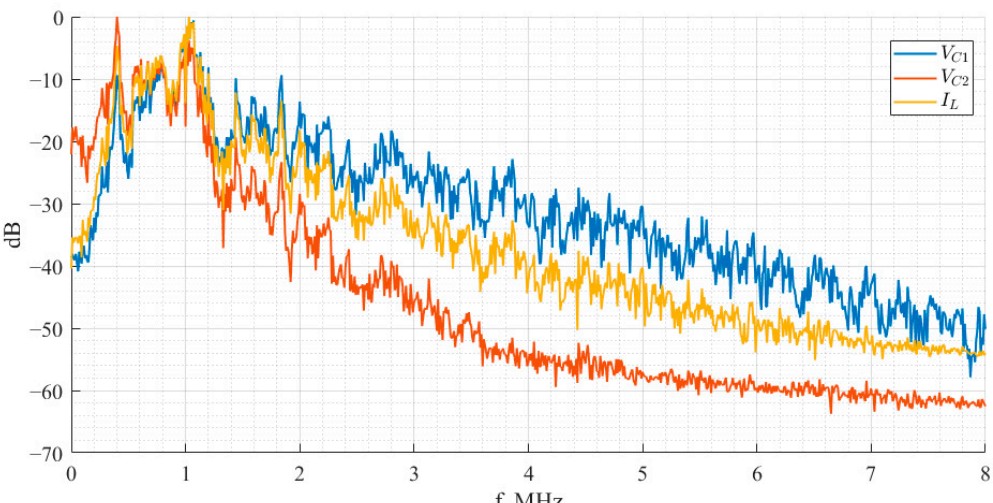

**Figure 5.** Colpitts Chaos oscillator signals $V_{C1}$ (blue), $V_{C2}$, (red) and $I_{L1}$ (yellow) spectra.

Furthermore, cross-correlation functions were calculated within a single Colpitts chaos oscillator to evaluate the chaotic behavior and determine the shortest bit length [59]. Figure 6 depicts the cross-correlation functions between $V_{C1}$ and $V_{C2}$, $V_{C1}$ and $I_{L1}$, and $V_{C2}$ and $I_{L1}$.

The examination of aperiodic signals (as chaotic signals) indicates a rapid reduction in the amplitude of the cross-correlation function as the time shift between signals grows (Figure 6), indicating the oscillator's complicated and non-repetitive dynamics. This unpredictability is especially useful for applications that require high randomness and complexity. The bit length is calculated by identifying the settling point of the amplitude

inside the cross-correlation function and linking it with the associated time shift. This method provides the proper choice of minimal bit length [59]. Based on the cross-correlation function analysis, a minimum bit length of 6 μs is required to ensure its distinguishability during detection, limiting the data transmission rate to 166.7 Kbps. Periodic signals, on the other hand, show a progressive decrease in the amplitude of the cross-correlation function with increasing time shift, indicating predictable and repeated activity inside the oscillator.

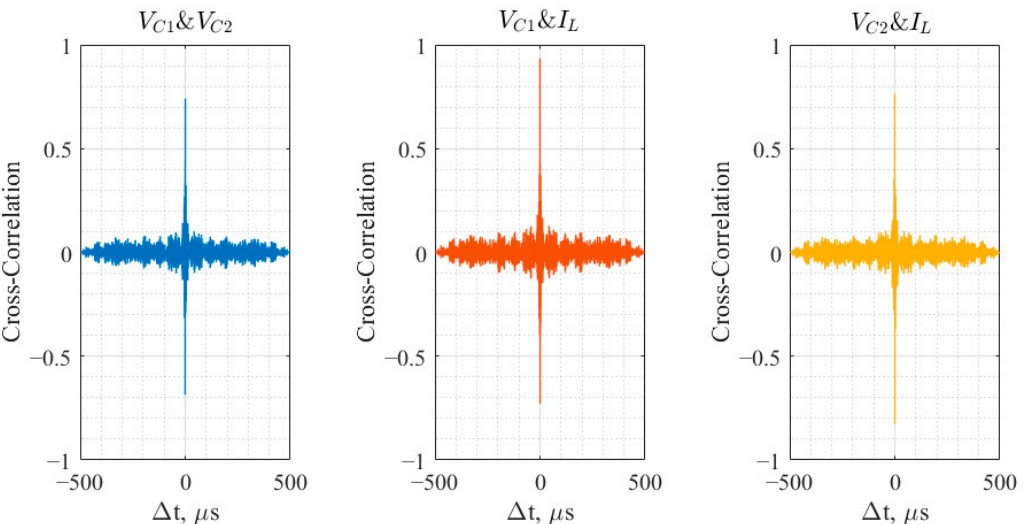

**Figure 6.** Cross-correlation functions between $V_{C1}$ and $V_{C2}$ (blue), $V_{C1}$ and $I_{L1}$ (red), and $V_{C2}$ and $I_{L1}$ (yellow).

### 3.3.2. Experimental Study

The Colpitts chaotic oscillator prototype (Figure 7a) was assembled and tested with Analog Discovery Pro. The chaos oscillator prototype was made using commercially available components with the same nominal element as in the simulation. As all components have nominal deviations and parasitic characteristics, the variable resistors were added to the circuit. Changing the resistance allows the oscillator to tune into chaotic behavior. A comparison of the two-dimensional projections of the attractor, shown in Figure 7b, with those from the original work [36], encountered coherence between the projections. This agreement assumes the prototype's successful completion. Minor differences in the projections are ascribed to the parasitic properties of the components. Furthermore, the Z1TEST result of 0.98 verifies the chaotic nature of the system.

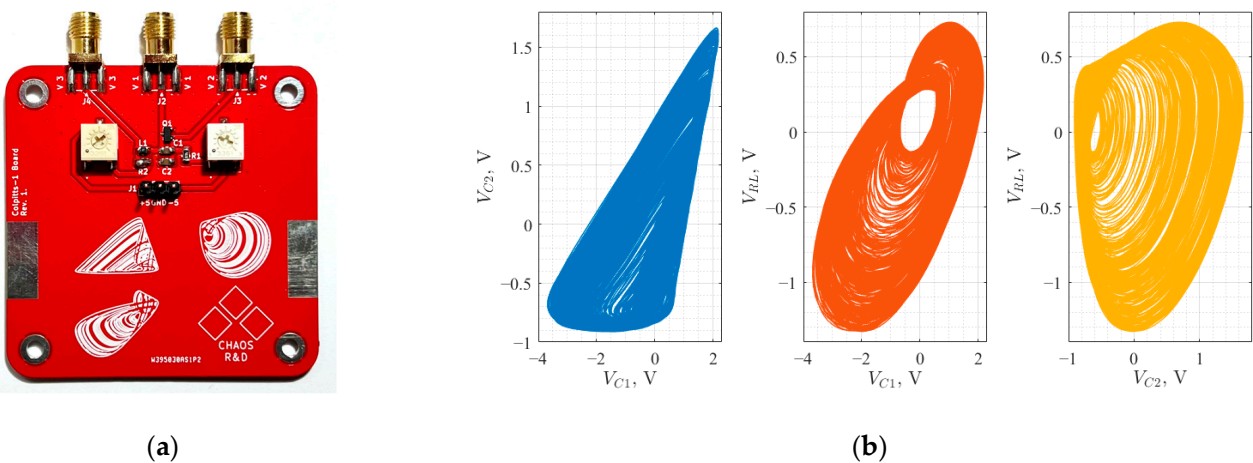

(**a**)

(**b**)

**Figure 7.** (**a**) Colpitts chaos oscillator prototype; (**b**) Colpitts chaos oscillator prototype's attractor two-dimensional projections.

The next goal was to synchronize two oscillators in the drive-response system. The substitution approach was used to create synchronization within the Colpitts' chaos oscillator via voltage across capacitors $C_1$ and $C_2$. Notably, correlation coefficient values in both configurations reached a significant 0.95, confirming the system's robust synchronization.

Figure 8 depicts the signal spectra for $V_{C1}$, $V_{C2}$, and $V_{RL}$. A close analysis of these spectra reveals a signal band centered predominantly on the chaos oscillator's fundamental frequency as well as a wide band, which is a property of the chaotic signal.

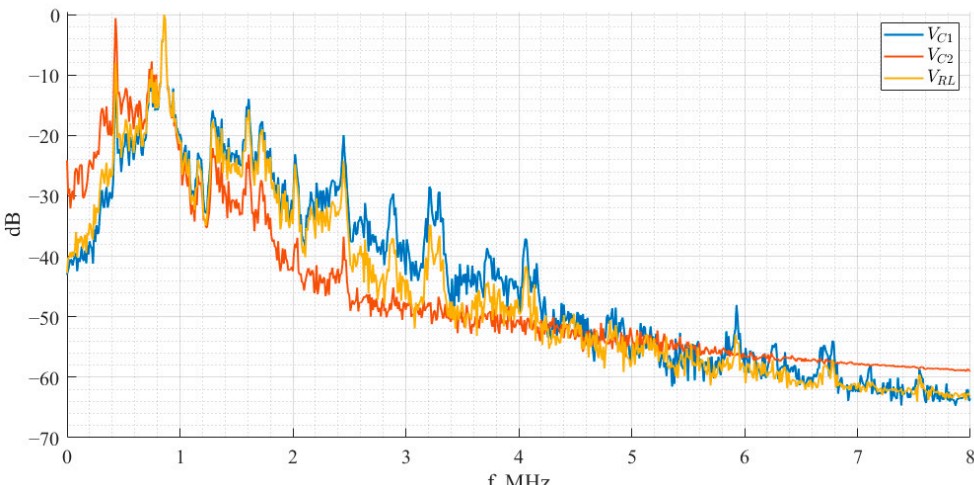

**Figure 8.** Colpitts Chaos oscillator signal $V_{C1}$ (blue), $V_{C2}$ (red) and $V_{RL}$ (yellow) spectra.

Likewise, cross-correlation functions were calculated within a Colpitts chaos oscillator prototype to evaluate its chaotic behavior and determine the minimal bit length. The cross-correlation functions between $V_{C1}$ and $V_{C2}$, $V_{C1}$ and $V_{RL}$, and $V_{C2}$ and $V_{RL}$ are shown in Figure 9.

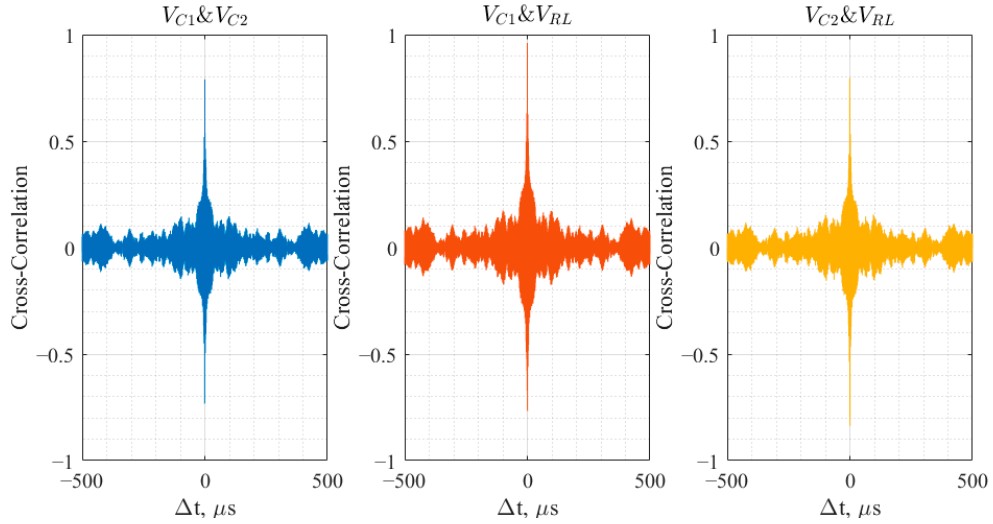

**Figure 9.** Cross-correlation functions between $V_{C1}$ and $V_{C2}$ (blue), $V_{C1}$ and $V_{RL}$ (red), and $V_{C2}$ and $V_{RL}$ (yellow).

The examination of aperiodic signals, notably chaotic signals from the prototype, shows a rapid decrease in the amplitude of the cross-correlation function as the time difference between signals grows (Figure 9). This behavior reflects the oscillator's complex and non-repetitive dynamics, emphasizing unpredictability. Based on the cross-correlation function analysis, a minimum bit length of 7.5 μs is required for successful data transmission, limiting a data transfer rate of 133.3 Kbps.

### 3.3.3. Simulation and Experimental Study Comparison

The comparison between simulation and prototype results enlightens critical aspects of the Colpitts chaos oscillator models. The attractor projections obtained from the simulation (Figure 4) and prototype measurements (Figure 7) align well, proving the agreement between the LTspice model and the real-world prototype. Differences between these projections are attributed to inherent imperfections within the electronic circuit components, contributing to minor deviations.

Examining the signal spectra in Figure 5 (simulation) and Figure 8 (prototype) reveals notable similarities, indicating a convergence in their fundamental frequency. However, a marginal disparity in noise levels is evident, with the prototype exhibiting slightly higher noise in its spectrum compared to the simulation. This discrepancy can be attributed to real-world conditions affecting the prototype's signal quality, a factor often absent or minimized in the controlled environment of simulations.

The cross-correlation functions depicted in Figure 6 (simulation) and Figure 9 (prototype) present a narrow central section with a rapid amplitude decline, characteristic of chaotic dynamics. However, discernible differences arise in the prototype results, where increased noise levels and higher oscillation amplitudes are observed. This variation leads to a more significant minimum bit length required for effective data transfer in the prototype compared to the simulation.

In summary, while the simulation and prototype align in fundamental behavior, slight discrepancies in noise levels, oscillation amplitudes, and the necessary bit length for data transmission highlight the influence of real-world conditions on the prototype's performance. These findings underscore the importance of considering practical imperfections and real-world influences when translating simulated models into physical prototypes.

### 4. The Quadrature Chaos Shift Keying Communication System's Performance Analysis

This section delves into the evaluation of noise immunity in the QCSK communication system (Figure 1, Section 2). The communication system is based on the Colpitts chaos oscillator, ensuring security on the physical layer due to the chaos signal properties. The system's model consists of a transmitter, an Additive White Gaussian Noise (AWGN) channel, and a receiver. Within the transmitter, a chaos oscillator operates in drive mode, employing the Chaos Shift Keying (CSK) scheme to form the information-carrying signal. The oscillator's state variables $X$ and $Y$ (depicted in Figure 1) distinctly map bit values '1' and '0', respectively. The bit rate of information is 128 kbps for future system employment in IoT systems.

The information-carrying and synchronization signals undergo modulation before transmission, simultaneously transmitting through the AWGN channel. These signals are demodulated and filtered at the receiver end using a low-pass filter. The filter was carefully adjusted with consideration for signal characteristics. The bandwidth was set to achieve a mean square error of $-40$ dB between the input and output signals. After filtration, ensuring synchronization of the response chaos oscillator is pivotal and achieved through the substitution method, leveraging the demodulated synchronization signal $Z'$. This synchronization step is fundamental for accurate and reliable signal detection.

Detection of a transmitted data bit relies on estimating correlation coefficients between the demodulated information-carrying signal and the state variables $X'$ and $Y'$ of the response chaos oscillator. Decision-making regarding the received bit is based on the calculated correlation coefficient.

This research findings indicate that synchronization within the Colpitts drive-response system can be established using $V_{C1}$ or $V_{C2}$. A communication system was developed for both scenarios to comprehensively study synchronization and the selection of information-forming signals for chaos-based data transmission systems. Table 1 provides a detailed breakdown of how signals were utilized in each study and the correlation coefficient between information-forming signals.

This study investigates the synchronization dynamics inside the drive-response system, considering the possible effects of employing $V_{C1}$ or $V_{C2}$ for synchronization. The correlation coefficient provides insight into the interaction and alignment of information-forming signals, aiding in the knowledge of signal selection for effective chaos-based data transfer. The purpose of an established communication system is to study the complex dynamics of the Colpitts chaotic oscillator in various synchronization setups.

**Table 1.** State Variables and Cross-Correlation Coefficient Between Information-Forming Signals.

| '1', X | '0', Y | Synchronization Signal, Z | Cross-Correlation Coefficient, X and Y |
|---|---|---|---|
| $V_{C2}$ | $V_{RL}$ | $V_{C1}$ | 0.11 |
| $V_{C1}$ | $V_{RL}$ | $V_{C2}$ | $-0.49$ |

*4.1. Simulation Study*

The QCSK data transmission system simulation comprises five essential blocks implemented in MATLAB. The MATLAB code scheme for studying the data transmission system is depicted in Figure 10.

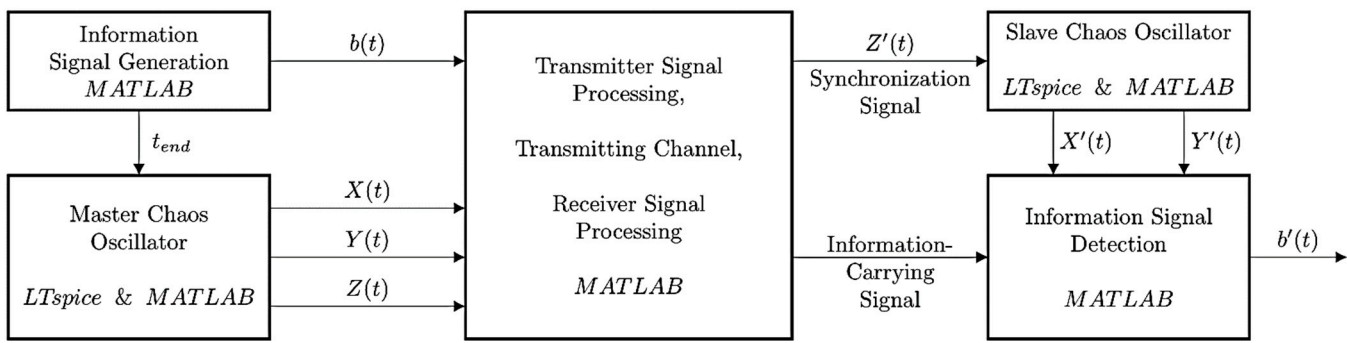

**Figure 10.** QCSK communication system simulation block-scheme, where $b(t)$ is information bit stream, $t_{end}$ is length of the drive chaos oscillator simulation, $X(t)$, $Y(t)$, and $Z(t)$ are drive chaos oscillator state variables, $Z'(t)$ is synchronization signal restored after signal processing in receiver, $X'(t)$ and $Y'(t)$ are state variables generated in response chaos oscillator, and $b'(t)$ is detected information stream.

The information signal is generated in the first block. Input parameters for the MATLAB code include bit length, number of bits, and sampling frequency. The simulation duration for the next step is calculated once the randomly generated bit sequence is saved to a text file. The subsequent step involves simulating the drive chaos oscillator, preserving three chaotic signals: one for chaotic synchronization and two for CSK. The simulation is carried out using the MATLAB function *LTspice2Matlab*(), which takes the required simulation netlist or code specifying the simulated scheme and then calls the LTspice software (Version 17.1.15) for the necessary simulation. The third stage involves signal processing prior to quadrature modulation, channel emulation, quadrature demodulation, and signal processing after demodulation. The response chaos oscillator is then simulated, with the synchronization signal from the previous signal processing block serving as input. The simulation is carried out with the help of the MATLAB function *LTspice2Matlab*(). The next step is to detect the information signal, which is accomplished by calculating the correlation coefficient within a sliding window. Consequently, the code outputs a BER (Bit Error Rate) curve.

*4.2. Experimental Study*

This study aims to assess the noise immunity of the prototype. The investigation involves examining the QCSK data transmission system prototype within an emulated AWGN channel, accomplished by incorporating an attenuator in the transmission path. Following the attenuation, the impact of noises on the QCSK communication system's

performance within the receiver unit became noticeable, prompting a thorough analysis of the QCSK data transmission system's noise immunity. The hardware study methodology scheme is shown in Figure 11.

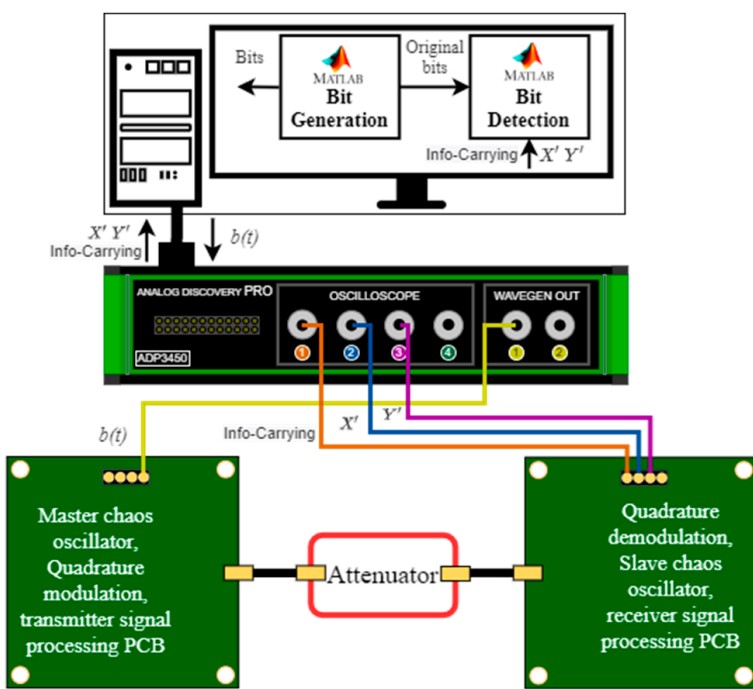

**Figure 11.** QCSK communication system hardware measurement block-scheme, where $b(t)$ is information bit stream, $X'$ and $Y'$ are state variables generated in response chaos oscillator.

The measurements were conducted using Analog Discovery Pro (ADPro), which serves the dual function of generating the information signal and capturing the information-carrying signal, and both signals from the response chaos oscillator correspond to bits '1' and '0'. Controlling the ADPro was achieved through Python code from a connected computer. The process of detecting the information signal was executed using custom MATLAB code. Subsequently, BER values were obtained from these experiments.

*4.3. Result Analysis*

Figure 12 shows the BER graphs for both synchronization signal scenarios within the communication system. The blue curve reflects the results achieved with $V_{C1}$ as a synchronization signal, whereas the red curve represents the results obtained with $V_{C2}$ as a synchronization signal.

When $V_{C1}$ was utilized as the synchronization signal, a BER of 10% was achieved for an $E_b/N_0$ of 8 dB; the curve demonstrates an almost linear descent with 2.4 powers of ten per decade for higher signal-to-noise ratio values. The red curve, on the other hand, depicts the noise immunity for the communication system that uses $V_{C2}$ synchronization. The graph demonstrates a linear drop of 0.5 powers of ten per decade within the $E_b/N_0$ ratio range of 20 dB to 30 dB. The curve's drop steepens to 1.8 powers of ten each decade between 30 dB and 40 dB.

The cross-correlation and spectrum properties of chaotic oscillator signals explain the variances in the obtained BER curves. The cross-correlation coefficient between the signals comprising the information-carrying signal is one of the components contributing to these results. The cross-correlation coefficient between $V_{C2}$ and $V_{RL}$ is 0.11, as indicated in Table 1, resulting in greater noise immunity compared to the second example. Increasing the cross-correlation coefficient's absolute value reduces system noise immunity. Understanding the

cross-correlation coefficient between signals that create information is critical, as it is the foundation of information detection.

The proposed QCSK communication system shows superior noise immunity in the AWGN channel compared to the FM-CSK [59], which achieves BER of 10% for the $E_b/N_0$ of 17 dB. Moreover, the QCSK communication system demonstrates better noise immunity than the Time reversal-differential chaos shift keying communication system (TR-DCSK) [63], achieving a BER of 10% at an Eb/N0 of 9 dB. In addition, the QCSK communication system has slightly better noise immunity than the QCPSK [64] communication system, which achieves BER of 10% for the $E_b/N_0$ of 8.5 dB. Furthermore, the QCSK data transmission system can achieve higher data transmission rates (up to 166.7 Kbps) compared to FM-CSK (up to 9 kbps) and QCPSK (up to 3 kbps), which shows the proposed system's suitability for IoT applications.

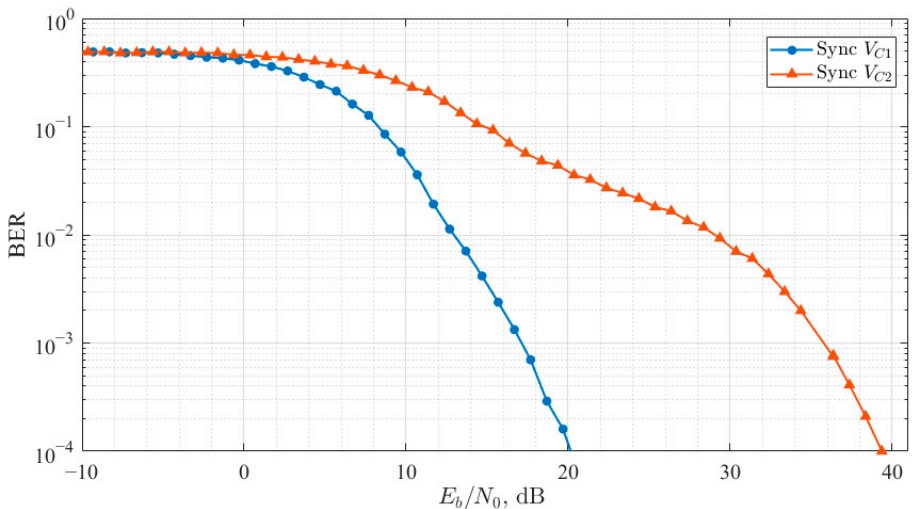

**Figure 12.** BER curves of QCSK communication system, where blue curve represents system's noise immunity, when $V_{C1}$ is employed as synchronization signal and red curve represents system's noise immunity when $V_{C2}$ is employed as synchronization signal.

## 5. Conclusions

This study aimed to explore the implementation and performance evaluation of a pioneering Quadrature Chaos Shift Keying (QCSK) chaotic communication system tailored for Internet of Things (IoT) applications. This innovative system leverages chaotic synchronization and quadrature modulation. The core component is the Colpitts chaotic oscillator operating within a drive-response configuration, enhancing the communication system's security at the physical layer. The Colpitts oscillator offers engaging characteristics such as a wide spectrum, noise-like waveform, narrow cross-correlation function, synchronization capability, and sensitivity to initial conditions, all while maintaining system determinism.

Comprehensive simulation and experimental studies were conducted to analyze the behavior of the Colpitts chaotic oscillator with a fundamental frequency of 96.86 kHz. The simulation was accomplished in the LTspice environment, while prototype measurements were performed using Analog Discovery Pro. Chaotic signals were processed in MATLAB for both studies. In the simulation, chaotic behavior was achieved using the same system parameters as in the original study, except reducing the values of $C_1$, $C_2$, and $L_1$ by 10 times to achieve a higher fundamental frequency. The design of the Colpitts chaos oscillator prototype was modified by adding two variable resistors for tuning. Both simulation and prototype results confirmed the characteristic chaotic dynamics of the Colpitts chaos oscillator, and synchronization in the drive-response system was successfully established. The comparison of signal spectra and construction of cross-correlation functions further validated the generation of chaotic signals, as confirmed by the Z1TEST.

The evaluation of noise immunity in the QCSK data transmission system within the AWGN channel revealed the influence of synchronization signal properties from the chaos oscillator on error probabilities. This study demonstrated superior performance for the QCSK communication system when VC1 was employed as the synchronization signal. Compared to previously explored TR-DCSK, FM-CSK, and QCPSK digital communication systems, the QCSK system, built on the Colpitts chaos oscillator, exhibited enhanced noise immunity during data transmission. Its simple design and improved security capabilities position this system as a promising solution for strengthening IoT communication security.

In the future, the investigation of the QCSK chaotic communication system can be expanded to practical applications in real-world IoT sensors. Deploying the system in real-world scenarios will offer vital information about its performance, resilience, and robustness. Experiments using real IoT devices will provide a more in-depth knowledge of the system's behavior in dynamic and diverse settings. Furthermore, additional optimizations and enhancements might be explored to improve the system's efficiency and handle any issues that arise during real-world deployments.

**Author Contributions:** Conceptualization, A.L.; Methodology, F.C. and D.K.; Software, R.B. and F.C.; Validation, D.C. and R.B.; Investigation, D.C.; Resources, R.B.; Writing—original draft, D.C.; Writing—review & editing, D.K. and A.L.; Supervision, D.K.; Funding acquisition, A.L. All authors have read and agreed to the published version of the manuscript.

**Funding:** This work has been supported by the European Social Fund within Project No. 8.2.2.0/20/I/008 "Strengthening of Ph.D. students and academic personnel of Riga Technical University and BA School of Business and Finance in the strategic fields of specialization" of the Specific Objective 8.2.2 "To Strengthen Academic Staff of Higher Education Institutions in Strategic Specialization Areas" of the Operational Program "Growth and Employment" and the RTU doctoral grant program.

**Institutional Review Board Statement:** Not applicable.

**Informed Consent Statement:** Not applicable.

**Data Availability Statement:** Publicly available datasets were analyzed in this study. This data can be found here: https://github.com/DarjaCirjulina/Experimental-Study-QCSK (accessed on 20 January 2024).

**Acknowledgments:** This research was performed at Riga Technical University, Space Electronics and Signal Processing Laboratory–SpacESPro Lab.

**Conflicts of Interest:** The authors declare no conflict of interest.

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
