# Peer review of "Experimental Study on Colpitts Chaotic Oscillator-Based Communication System Application for the Internet of Things"

_applsci, doi:10.3390/app14031180_

Round 1

Reviewer 1 Report

Comments and Suggestions for Authors

Authors have conducted a comprehensive scientific study, spanning from simulation to implementation and testing. The paper is well-structured and of high quality. To further enhance its quality, I have provided some suggestions below that may prove beneficial for the authors.

Line 160: The authors introduce state variables as 'voltages across capacitors and currents through inductors' without establishing a clear rationale for defining these variables in physical terms at this juncture of the paper. Additionally, this wording appears peculiar and lacks alignment with Figure 1.

Line 242: The correlation coefficient is introduced; however, it is not specified whether it pertains to the Pearson correlation coefficient. Clarification on the type of correlation coefficient, especially if it is Pearson, would enhance precision in the paper.

Line 251: The term 'voltage repeater' in the context of synchronization appears unclear and might be a typographical error or misunderstanding. Consider providing clarification on whether 'voltage repeater' is accurate or if 'follower' was intended. Additionally, including a schematic or more detailed explanation would bolster the claim and enhance clarity in the paper.

Line 259: Please explain what Z1TEST is.

Line 311: Clarifying why a 6 μs bit length is deemed 'required' would benefit the reader's understanding. Consider providing a rationale for this specific bit length, such as its relevance to the discussed method's objectives or how it aligns with data rates in NB-IoT. It may be helpful to introduce the NB-IoT data rate first and then explain how the chosen bit length contributes to achieving similar characteristics. This would provide a clearer context for readers to comprehend the significance of the specified parameters.

Figure 6: Adding a zoomed-in figure around t=0, focusing on a dozen microseconds, could indeed enhance the readers' understanding of the cross-correlation characteristics when its value is significant. This would provide a detailed view of the critical region and help readers grasp the nuances of the correlation dynamics during that period. It's a thoughtful suggestion to improve the clarity and detailed representation of the data, especially in relation to the crucial 6 μs bit length.

Figure 7: Both a) and b) has poor quality. Could photo have slightly better resolution? The improved resolution of attractors would provide better clarity, preventing the lines from merging into a single stain of color. Additionally, a shorter time of observation might reveal the chaotic characteristics more distinctly, allowing for a clearer representation of the attractors. These adjustments should contribute to a better visualization of the chaotic behavior in the system.

Figure 9: Providing a zoomed-in view of a narrower area, specifically around a dozen microseconds, in a separate figure (similar to Figure 9) would allow readers to better observe and understand the behavior of interest. This would enhance the clarity and detail of the representation, aiding in the interpretation of the data for that specific time range.

Figure 12: Ensuring accessibility for visually impaired readers and those who may read the paper in grayscale is a commendable consideration. To address this, you may want to use different line styles or patterns (e.g., solid, dashed, dotted) in addition to colors. This will enhance the distinction between lines, making the figures more accessible for a wider range of readers, including those with color vision deficiencies. This remark may be also applied to figures 3, 5, 8 although with lesser extent.

In conclusion, the paper is highly engaging, well-crafted, and I eagerly anticipate its publication.

Comments on the Quality of English Language

Line 124 probably has typo: 'enchanting' instead of 'enhancing'.

In the paper, the terms 'master,' 'slave,' and 'master-slave' are employed. However, these terms carry historical connotations and may not precisely convey the dual nature of technical systems. Contemporary engineering language tends to favor more neutral pairs such as 'primary/secondary' (and 'tertiary'), 'local/remote,' 'transmitter/receiver,' 'main/subsidiary,' 'principal/subordinate,' and 'cardinal/ancillary.' In Figure 1 of the paper, the pair 'transmitter/receiver' is used. Please consider adopting these alternatives for a more inclusive and updated terminology, ensuring consistency throughout the text.

Author Response

  1. Line 160: Thank you for your insightful comments on Line 160. We appreciate your suggestion for clarification, and we have incorporated the information about our system being based on analog oscillators. We aim to enhance the reader's understanding and facilitate easy referencing throughout the paper by introducing the connection between circuit theory, differential equations, and state variables.
  2. Line 242: Your input has been valuable, and we have addressed this concern by adding necessary clarification to the manuscript.
  3. Line 251: We appreciate your attention to detail on Line 251 and your suggestion to change the 'voltage repeater' to 'voltage follower.' 
  4. Line 259: We acknowledge the need for additional clarity in explaining the Z1TEST process. We have included a detailed description of the four primary steps involved, ensuring a better understanding for our readers.
  5. Line 311: Your suggestion to improve the cross-correlation function description and desirable transmission speed justification have been implemented, enhancing the overall coherence of our manuscript. We made clarifications in the manuscript in the cross-correlation function description. We also moved our motivation and desired transmission speed to the introduction.
  6. Figures 6 and 9: Your attention to the importance of the critical region close to t=0 in Figures 6 and 9 is highly appreciated. While we agree with the significance of this aspect, we also emphasize the broader picture for a comprehensive understanding of the system's non-repetitive behavior in chaotic scenarios.
  7. Figure 7 (a) and (b): Thank you for your feedback on Figure 7. We acknowledge the impact of PDF conversion on the resolution of the photos and graphs. We will provide the original pictures and graphs with the resubmitted manuscript to address this, ensuring optimal clarity.
  8. Figure 12: We are grateful for your consideration of accessibility in Figure 12. Your suggestion to modify the figure for visually impaired readers and those viewing the paper in grayscale has been implemented to ensure inclusivity and accessibility.
  9. Line 124: Thank you for catching the typo on Line 124. The term 'enchanting' has been corrected to 'enhancing' in the manuscript.
  10. Terminology Change: We are grateful for your suggestion to replace the terms 'master' and 'slave'. We replaced the terms throughout the manuscript with 'drive' and 'response'.

We truly appreciate the time and effort you invested in reviewing our manuscript. Your valuable insights have significantly contributed to its improvement. If there are any further suggestions or concerns, please do not hesitate to let us know.

Reviewer 2 Report

Comments and Suggestions for Authors

This work presents an experimental study of Quadrature Chaos Shift Keying (QCSK) as a means to tighten the physical layer security of IoT communication.

Overall, this idea is interesting and it is good to see more practical assessment on this topic. Some aspects can be improved.

- the motivation is not very clear, suggest the authors can separate Introduction with sub-part, e.g., 1.1, 1.2...

- the state-of-the-art can be summarized in a table

- a comparison with similar studies should be given

- how to set up the experimental environment can be discussed, what will be the main technical challenges?

Comments on the Quality of English Language

overall is okay, but can do a double check on syntax errors

Author Response

  1. We genuinely appreciate the reviewer's comments on the introduction structure. While we value the suggestion to divide the introduction into subsections, we believe that maintaining its current format enhances readability and reduces potential confusion for the reader. We have carefully considered your input and opted to retain the current organization to ensure a smooth and cohesive flow throughout the manuscript.
  2. Thank you for your valuable feedback regarding summarizing different systems in one table. We understand the challenges posed by diverse metrics used by different authors. Creating a comprehensive table may not be feasible due to this variability. However, we have taken your suggestion into account and will strive to present the information in a clear and accessible manner within the limitations posed by the diverse metrics employed.
  3. To address your suggestion, we have extended our comparison in the result analysis and the introduction part. This expansion aims to provide a more thorough and comprehensive understanding of the context for our readers.
  4. We have considered your feedback and modified the manuscript to emphasize better the main challenge. Tuning the chaotic oscillator to achieve chaotic behavior during experimental measurements is the main technical challenge. 

We genuinely appreciate the time and effort you dedicated to reviewing our manuscript. Your feedback has been invaluable in refining our work.

Reviewer 3 Report

Comments and Suggestions for Authors

In this manuscript, the authors present an experimental study of QCSK. The QCSK communication system is built using an analog chaos oscillator. The ability to synchronize and low signal cross-correlation of the analog chaos oscillator were studied by spice simulation, numerical simulation, and circuit board experiments. The experimental results match the simulation results well. However, certain issues need to be addressed before considering it for publication.

1. Does the chaos oscillator system possess control parameters? Within what parameter range is the system chaotic?

2. Bifurcations and Lyapunov exponents are commonly used for analyzing chaotic systems. The authors should explain the reasons for choosing the 0-1 test method to evaluate the system's chaotic behavior.

3. Another obvious problem with this paper is the lack of sufficient explanation of the simulation results. You need to explain your simulation results in detail and why you got such results.

4. Conclusions need more in them, as they're more of an afterthought. The authors are suggested to highlight important findings and include an afterthought of this work.

Author Response

  1. We appreciate your inquiry about control parameters in the chaos oscillator system and the range within which the system exhibits chaotic behavior. In the physical system, all components impact the system's dynamic, because of the nominal deviations and parasitic characteristics. The simplest way to tune the oscillator to the chaotic dynamics is to use a variable resistor. Typically, it is hard to predict the range because each prototype would be different due to the nominal deviations of the commercially available components. To provide more clarity, we have added information on this aspect in the revised manuscript.
  2. We acknowledge the common use of bifurcations and Lyapunov exponents in analyzing chaotic systems. Our decision to opt for the 0-1 test method is rooted in its simplicity and suitability for discretized signal analysis, which aligns with our specific need to evaluate signals in our study. On the other hand, bifurcations and Lyapunov exponent are used to analyze the system. Also, studies on the Colpitts chaos oscillator bifurcation and Lyapunov exponent analyses were conducted by other authors.
  3. We have revised the manuscript to offer a comprehensive explanation of our simulation results, addressing the specific reasons behind the obtained outcomes. This enhancement ensures a more thorough understanding for our readers.
  4. We recognize the need to enhance the Conclusions section to highlight significant findings better and provide a more thoughtful reflection on the significance of our work. The conclusions have been modified to incorporate a more in-depth analysis and thoughtful afterthought, enriching the overall impact of our study.

We genuinely appreciate your thoughtful questions and the opportunity to address these important aspects of our manuscript. If there are any further points or concerns, please feel free to let us know.

Reviewer 4 Report

Comments and Suggestions for Authors

- The challenges and achievements of the work should be highlighted.

- The authors mentioned that 'The proposed QCSK communication system shows superior noise immunity in the AWGN channel compared to the FM-CSK and slightly better noise immunity than the QCPSK communication system'.  How? Where is the simulations to verify this superiority?

- What kind of research can be directed in the future for this study?

There are some typographical errors in the manuscript. For example, please check reference [49]; the authors are missing.

Author Response

  1. We appreciate the suggestion to highlight the challenges and achievements of our work. To address this, we have added relevant information in the experiment description and modified the conclusions section. By doing so, we aim to provide readers with a clearer understanding of the difficulties encountered and the accomplishments achieved in our research.
  2. We have included specific results with relevant numbers to substantiate our claim. These additions offer a more concrete basis for asserting the system's noise immunity compared to FM-CSK and QCPSK in the AWGN channel. Simulations of FM-CSK and QCPSK in the AWGN channel were presented in cited papers.
  3. Considering future research directions, we have incorporated information on possible methods in the conclusions section.
  4. We acknowledge the identification of typographical errors, especially in reference [49]. We have thoroughly checked and corrected the reference, including all necessary author information. Additionally, we have reviewed the entire manuscript and addressed various typographical errors for improved overall quality.

We sincerely appreciate your detailed feedback and constructive suggestions. If there are any further points or concerns, please feel free to communicate them.

Reviewer 5 Report

Comments and Suggestions for Authors

While the overall technical merits of the paper are fine, the Introduction needs to be corrected at a large scale: there are various cryptography related errors, especially when it comes to terminology.

See, e.g., line 71 (SHA256 is a hash function!). Also, I suggest to talk less about blockchain as it is not that important compared to other crypto-related mechanisms, especially in the IoT context.

In case the authors have a cryptographer in their insitution, I suggest asking for some technical help, as the understanding problems regarding concepts seem obvious.

I suggest to improve the work with some newer, meaningful references at least on the crypto side. Also, I see room for improvement regarding the motivation.

Comments on the Quality of English Language

Please use a spell and grammar checker, I spotted a few typos and extra spaces between words (including the Abstract).

Author Response

We appreciate the thorough review and constructive feedback provided by the reviewer. We have carefully corrected terminology in response to concerns about cryptography-related errors in the introduction.

We value the suggestion to include newer references and improve motivation, even though cryptography is not the primary focus of our research; we have incorporated recent publications to represent this aspect more effectively and revisited the motivation section for a more compelling rationale. We thank the reviewer for their valuable insights, which have significantly contributed to refining our manuscript's accuracy and relevance of cryptographic concepts.

Round 2

Reviewer 4 Report

Comments and Suggestions for Authors

The paper can be accepted now.

Reviewer 5 Report

Comments and Suggestions for Authors

The authors have at least partially addressed my comments. Even though I fully agree that cryptography should not be the main focus of the paper, my comments were related with the fact that incorrect/incomplete/out of the scope facts do not have to be included in a serious scientific paper.